# Federated Disentangled Tuning with Textual Prior Decoupling and Visual Dynamic Adaptation

Yihao Yang [* 1]   Wenke Huang [* 1]   Guancheng Wan [* 1]   Bin Yang [1]   Mang Ye [1]

## Abstract

Federated Parameter-Efficient Fine-Tuning aims to adapt Vision-Language Models for downstream tasks in distributed environments. However, data heterogeneity across participants hinders collaborative effectiveness, necessitating personalized adaptation to cover distinct data distributions. Current personalized methods suffer from two limitations. 1) Textual Property Loss: Existing methods facilitate the collaboration between decoupled prompts at the feature level, which potentially undermines the textual properties of the prompts. 2) Visual Feature Diversity: The diversity of visual features makes it challenging to leverage naive image features directly for image-text alignment in downstream tasks. In this work, we propose **Fed**erated **D**isentangled Tuning with Textual Prior **D**ecoupling and Visual Dynamic **A**daptation (FedDDA) to overcome the above limitations. Specifically, we encourage decoupling prompts in a way that maximizes the efficacy of prior knowledge, which is essential for maintaining a coherent linguistic context. Furthermore, we design a visual adaption model to reshape visual space to optimally align with the textual space. Extensive experiments on various image classification tasks show the effectiveness of our work in addressing data heterogeneity. The codes are released at `https://github.com/MoratalYang/FedDDA`.

## 1. Introduction

Federated learning is a distributed paradigm (McMahan et al., 2017; Konečný et al., 2016a;b; Mohassel & Zhang,

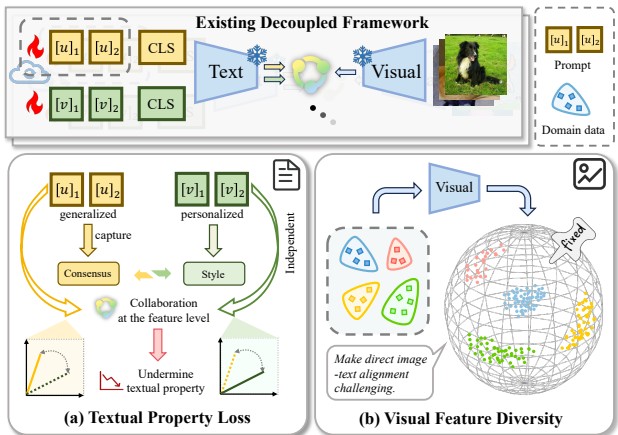

*Figure 1.* **Background and motivation.** The **top** row outlines the framework for existing decoupled methods. The **bottom** row illustrates our motivation in both modalities. (a) In popular decoupled methods, the collaboration between generalized and personalized prompts relies on newly loss signals at the feature level, which undermines the textual properties of the prompts. (b) The diversity of visual features, stemming from the complex information within images, makes direct image-text alignment challenging.

2017; Yang et al., 2019), allowing collaborative model training across decentralized clients without exposing their local data. Recently, pre-trained Vision-Language Models (VLMs), *e.g.*, CLIP (Radford et al., 2021) and ALIGN (Jia et al., 2021), have demonstrated exceptional versatility across a wide range of downstream tasks. Meanwhile, Parameter-Efficient Fine-Tuning (PEFT) (Xu et al., 2023; Luo et al., 2023) acts as a promising solution to adapt to the VLM, which updates a limited number of parameters while keeping the remaining ones frozen. Leveraging its lightweight nature, Federated Parameter-Efficient Fine-Tuning (FedPEFT) has emerged as a privacy-friendly collaboration paradigm, enabling distributed data sources to jointly fine-tune VLMs. Several studies (Guo et al., 2023b; Lu et al., 2023; Qiu et al., 2024; Feng et al., 2023) follow the FedAvg paradigm (McMahan et al., 2017) to aggregate selected candidate parameters.

However, similar to conventional FL, FedPEFT faces challenges posed by data heterogeneity (Zhao et al., 2018; Kairouz et al., 2021; Li et al., 2021a; 2020a), which is inherent in distributed environments. Specifically, data from mul-

[*]Equal contribution [1]National Engineering Research Center for Multimedia Software, School of Computer Science, Wuhan University, Wuhan, China. Correspondence to: Mang Ye <ye-mang@whu.edu.cn>.

*Proceedings of the 42nd International Conference on Machine Learning*, Vancouver, Canada. PMLR 267, 2025. Copyright 2025 by the author(s).

tiple clients typically presents Non-IID (non-independent and identically distributed) distributions. Therefore, solely learning the shared parameters brings the restricted representations to cover distinct distributions. Personalized Fed-PEFT (Su et al., 2024; Guo et al., 2023a) acts as a crucial role to benefit the multiple collaboration under data heterogeneity. For personalized adaptation, decoupling is key to achieving fine-grained representations. Recent methods (Bai et al., 2024; Cui et al., 2024; Li et al., 2024) follow the decoupling strategy by constructing independent prompts with specific properties, *e.g*., generalization and personalization, and designing external signals to guide optimization.

However, as illustrated in Fig. 1, these decoupled researches are baffled by two certain limitations stemming from their paradigm. **I) Textual Property Loss**: With respect to textual modality, these methods rely on newly designed loss signals to facilitate collaboration among independent prompts at the feature level. However, this collaborative mode does not explicitly decouple prompts at the semantic level, potentially undermining the textual properties of the prompts. Specifically, the textual properties are shaped by prior knowledge acquired through extensive image-text alignments, which plays a vital role in constructing a coherent semantic. Yet, independent prompts derive minimal benefits from prior knowledge, as they are fed into models separately, while prior knowledge is typically built upon a single input. Thus, our goal is to achieve better textual decoupling in a manner that maximizes the efficacy of prior knowledge. **II) Visual Feature Diversity**: Towards the visual modality, a crucial issue is the diversity of vision, which is overlooked by previous personalized methods. Unlike uniform labels, images within the same category often exhibit varied styles, leading to a diverse visual feature space. Additionally, the mode of image-text alignment depends on the distribution of both visual and textual latent spaces (Cho et al., 2023a). While the textual space can be optimized by textual prompts, the diverse visual space makes it challenging to leverage fixed features directly for image-text alignments. Hence, it's crucial to optimize visual space that is optimally aligned with the textual space.

To address these issues, we propose a simple yet effective algorithm, **Fed**erated **D**isentangled Tuning with Textual Prior **D**ecoupling and Visual Dynamic **A**daptation (FedDDA). For **I**), we introduce Textual Prior Decoupling (TPD) to decouple generalized and personalized prompts. Specifically, we rethink the utilization of hand-crafted prompts. As highlighted in (Cho et al., 2023b; Gal et al., 2023), hand-crafted prompts with domain-specific labels provide greater discrimination across multi-domain scenarios by leveraging abundant prior knowledge. This motivates us to integrate generalized representations with personalized ones within a robust linguistic context. Specifically, we explicitly decouple the prompts into global and local components connected

by guidance words, with each responsible for extracting consensus and style knowledge, respectively. In this way, the semantic fusion of consensus and style knowledge draws support from prior knowledge without additional signals. In response to **II**), we propose Visual Dynamic Adaptation (VDA) to reshape the visual space in a dynamic manner. Specifically, we introduce a dual adapter architecture to decouple consensus and style features from naive visual features and subsequently construct an adaptive visual representation. Given the federated setting, one adapter is shared while the other is private. The shared adapter extracts client-invariant visual information, *i.e*., consensus knowledge, while the private adapter captures client-specific visual information, *i.e*., style knowledge. Inspired by Mixture-of-Experts (MoE) (Shazeer et al., 2017; Masoudnia & Ebrahimpour, 2014; Cai et al., 2024; Chen et al., 2023), we further implement a gating mechanism to dynamically harmonize visual representations. Based on the consensus and style knowledge, this visual adaptation module can adaptively learn the optimal visual space that aligns with textual space.

In this paper, our work combines Textual Prior Decoupling with Visual Dynamic Adaptation to achieve decoupling from both textual and visual modalities. Textual Prior Decoupling learns fine-grained representations for robust language supervision, while Visual Dynamic Adaptation (VDA) adjusts visual features to optimally align with the textual space. In a nutshell, the main contributions are as follows:

- We focus on Personalized Federated Parameter-Efficient Fine-Tuning, highlighting two limitations of existing works from both textual and visual modality.
- We propose a simple yet effective method FedDDA. Through Textual Prior Decoupling and Visual Dynamic Adaptation, FedDDA obtains effective personalized vision-language models.
- We conduct extensive experiments on four datasets: Office31, PACS, OfficeHome, and DomainNet. Through a set of ablation studies, we validate the efficacy of FedDDA and the indispensability of each module.

## 2. Related Work

### 2.1. Parameter-Efficient Fine-Tuning

As model parameter counts increase, fine-tuning the entire model becomes prohibitively expensive and impractical. Parameter-Efficient Fine-Tuning (PEFT) (Xu et al., 2023; Luo et al., 2023; Huang et al., 2025b;a) has been introduced to address this challenge by reducing the number of active parameters while maintaining comparable capability to the full fine-tuning. Specifically, PEFT involves updating only a limited number of additional parameters or selectively optimizing a subset of the original parameters for efficient adaptation. Instead of selecting a subset, adding new parameters adopts a plug-and-play approach, keeping the

original model parameters frozen, such as reparameterized fine-tuning (Valipour et al., 2023; Xu et al., 2024) and additive fine-tuning (Zhou et al., 2022b; Gao et al., 2024). For instance, Low-Rank Adaptation (Hu et al., 2021) (LoRA) is the foundational work of reparameterized fine-tuning, where trainable low-rank decomposition matrices are injected into the network. Additive fine-tuning encompasses two primary techniques: prompt-based tuning and adapter-based tuning. Prompt-based tuning (Zhou et al., 2022a; Sun et al., 2022; Khattak et al., 2023; Yao et al., 2023; Zhang et al., 2024b; Fang et al., 2025) employs a set of contextual vectors that are inserted either into the input embeddings alone or across all intermediate layers, to stimulate task-specific knowledge. Adapter (Gao et al., 2024; Zhang et al., 2024a; 2022), actually a lightweight adaptation model at the end of the encoder, reshapes generated features into a new space, which can also be inserted into each layer. In this paper, we focus on PEFT based on VLMs and integrate textual prompts with visual adapters to establish a complementary relationship between language and vision.

## 2.2. Data Heterogeneous Federated Learning

With growing privacy concerns, Federated Learning (FL) (McMahan et al., 2017; Konečný et al., 2016a;b; Mohassel & Zhang, 2017; Yang et al., 2019; Ye et al., 2025) has been proposed as a framework for collaboratively training models across decentralized clients without leaking sensitive data. As a milestone, FedAvg (McMahan et al., 2017) updates a global model by aggregating local parameters from multiple clients and subsequently sends the updated model back for further training. However, FedAvg struggles with challenges posed by Non-IID data distributions (Zhao et al., 2018; Kairouz et al., 2021; Li et al., 2021a; 2020a; Ma et al., 2025) (*i.e.* data heterogeneity), which can negatively impair model performance. A common form of heterogeneity is label shifts, arising from imbalanced sampling. Several methods (Li et al., 2020b; Acar et al., 2021; Li et al., 2021b; Tan et al., 2022; Hu et al., 2024; Karimireddy et al., 2020) leverage global penalty term to address this challenge. However, these methods focus on single domain performance and overlook domain shift, where data from diverse sources inevitably falls into distinct feature distributions. This discrepancy leads to optimization inconsistencies between clients and the central server. Some efforts (Li et al., 2021c; Liu et al., 2021; Wu & Gong, 2021; Ye et al., 2023; Huang et al., 2023; 2022) attempt to develop personalized models for these multi-domain scenarios. However, the above methods mainly target conventional networks. In this paper, we explore a novel federated paradigm based on the vision-language model, *e.g.*, CLIP, which leverages contrastive learning to construct image-text alignment. This paradigm does not require training from scratch on closed datasets, enabling rapid adaptation.

## 2.3. Personalized Federated Parameter-Efficient Fine-Tuning

Federated Parameter-Efficient Fine-Tuning integrates PEFT techniques into federated learning to reduce communication and computational burdens. Some efforts (Guo et al., 2023b; Lu et al., 2023; Qiu et al., 2024; Feng et al., 2023) focus on shared parameters across different PEFT architectures to achieve generalized capability, similar to FedAvg (McMahan et al., 2017). For instance, PromptFL (Guo et al., 2023b) extends prompt tuning (Zhou et al., 2022b) to the federated setting by utilizing a set of shared learnable context vectors to activate prior knowledge. Similarly, FedCLIP (Lu et al., 2023) adopts a shared attention-based adapter within the visual branch to identify task-specific features. However, direct aggregation in these methods encounters the same impediments as conventional FL due to data heterogeneity. To address this issue, Personalized Federated Parameter-Efficient Fine-Tuning has been proposed to accommodate diverse distributions. Some methods introduce private parameters tailored to individual clients while maintaining shared prompts to foster global knowledge, such as pFedPrompt (Guo et al., 2023a) and FedAPT (Su et al., 2024). Moreover, approaches like FedOTP (Li et al., 2024), FedPGP (Cui et al., 2024), and DiPrompT (Bai et al., 2024) decouple prompts into independent components with distinct properties. These properties are typically derived from global or local federated setups and are further enhanced through external loss signals or multi-stage training. In this paper, we analyze the limitations of existing decoupled works from both modalities and further explore the decoupling strategy.

## 3. Methodology

In this section, we present the details of our proposed Fed-DDA, as depicted in Fig. 2. We leverage Textual Prior Decoupling (Sec. 3.2) to learn fine-grained representations that effectively capture both consensus and style knowledge, and employ Visual Dynamic Adaptation (Sec. 3.3) to reshape visual features in a dynamic manner. After these improvements, we follow the standard CLIP loss to achieve language-vision collaboration (Sec. 3.4).

## 3.1. Preliminary

**CLIP and Parameter-Efficient Fine-Tuning.** By aligning 400 million image-text pairs, CLIP obtains remarkable zero-shot capabilities, enabling generalized performance across diverse tasks. However, this generalized performance falls short for specific tasks. To enhance adaptability to downstream tasks, researchers have introduced Parameter-Efficient Fine-Tuning (PEFT), which fine-tunes the decision boundaries of VLMs at minimal cost by updating only a limited set of parameters. In this paper, we employ two typical

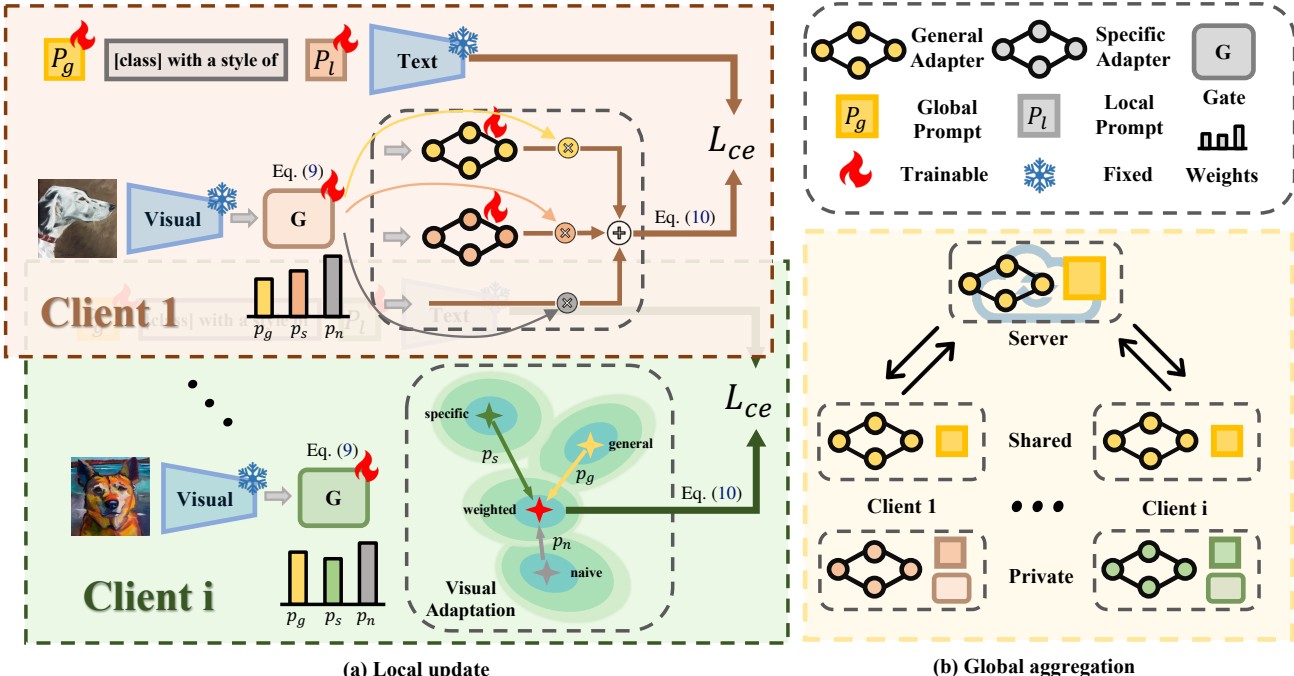

**(a) Local update**

**(b) Global aggregation**

*Figure 2.* **Architecture Illustration** of the FedDDA. In Textual Prior Decoupling (Sec. 3.2), we explicitly decouple the prompts into global and local components connected by guidance words, with each responsible for extracting consensus and style knowledge, respectively. In Visual Dynamic Adaptation (Sec. 3.3), dual adapters with gate deconstruct visual features and subsequently construct an adaptive visual representation in a dynamic manner.

PEFT techniques: prompt and adapter, which incorporate additional parameters into the models. For clarity, we use prompt tuning to illustrate the workflow of PEFT in this section, while details regarding the adapter can be viewed in Sec. 3.3.

Prompt tuning learns a set of vectors $p$, denoted as $\{p_1, \ldots, p_L\}$. Specifically, the learnable vectors are combined with class token embedding $c$. The input tokens of $k$-th class $P_k$ can be formulated as $\{p_1, \ldots, p_L, c_k\}$. Let $I(\cdot)$ denote the image encoder and $T(\cdot)$ denote the text encoder. Given an image sample $x$ with label $k$, the prediction probabilities can be calculated as:

$$q(k|x) = \frac{\exp(sim(I(x), T(P_k))/\tau)}{\sum_{c=1}^{K} \exp(sim(I(x), T(P_c))/\tau)}, \quad (1)$$

where $sim(\cdot, \cdot)$ denotes cosine similarity, $\tau$ is the temperature of Softmax and $K$ is the number of class. After calculating the probability, additional parameters $\theta$ (*i.e.* $p$ in prompt tuning) are optimized through contrastive learning while freezing the original ones:

$$\mathcal{L}_{ce}(x, y) = -\sum_{k=1}^{K} y \log q(k|x), \quad (2)$$

where $y$ is a one-hot label vector and $\mathcal{L}_{ce}$ represents the cross-entropy loss.

**Personalized Federated PEFT.** Considering a federated scenario with $M$ clients and a central server, each client $i$ is equipped with a CLIP model and holds a local dataset $D_i = \{(x_j, y_j)\}_{j=1}^{m_i}$ with $m_i$ scale. Denote $D = \{D_1, D_2, \cdots, D_M\}$ as the entire collection of datasets from diverse domain and $C_t$ as the set of active clients. During local training, the optimization objective is to minimize the cross-entropy loss in Eq. (2):

$$\arg\min_{\theta_i} \mathbb{E}_{(x,y)\sim D_i} \mathcal{L}_{ce}(x, y), \quad (3)$$

To minimize the prejudice of aggregated parameters on personalization, private parameters are introduced. Specifically, the updated parameters $\theta_i$ consist of global and local components, denoted as $\{\theta_{g,i}, \theta_{l,i}\}$. During the collaborative phase, each client transmits the generalized parameters $\theta_{g,i}^{t-1}$ to the server, where they are aggregated with weights proportional to the scale of the client's data. The updated global parameters for communication round $t$ is:

$$\theta_g^t = \sum_{i \in C_t} \frac{m_i}{\sum_{j \in C_t} m_j} \theta_{g,i}^{t-1}. \quad (4)$$

Subsequently, the server distributes the aggregated parameters back to the clients for further training.

### 3.2. Textual Prior Decoupling

**Motivation.** Driven by the decoupling strategy, we tend to separate the prompts for fine-grained representation. Specif-

ically, we seek to integrate consensus and style without compromising semantic integrity. This fusion can be effectively achieved by prior knowledge, which facilitates the nuanced understanding of the relationships between input tokens. A hand-crafted prompt that combines consensus and style, *e.g.* "a dog with a style of cartoon", is the most straightforward way to establish the cooperation between consensus and style representations, with the aid of prior knowledge. Consequently, we aim to decouple generalized and personalized prompts as consensus-and-style texts, leveraging prior knowledge to create a robust linguistic context.

**Decoupled Textual Prompts.** Specifically, we explicitly decouple the prompt into a global generalized prompt $P_g$ and a local personalized prompt $P_l$, both of which consist of a set of continuous vectors $\{p_1, ..., p_L\}$. To preserve textual semantics, these two prompts are combined into a single prompt, denoted as $P_i$. With respect to the text modality, the text encoder has the ability to embed the textual style guidance, due to its exposure to a broad range of styles when pre-training. Building on this insight, we introduce guidance words, "with a style of", between two prompts to evoke stylistic semantics. Hereby, we define $P_i$ as:

$$P_g \ [\text{class}] \text{ with a style of } \ P_l \ , \tag{5}$$

where $P_g$ is sent to the server for aggregation, capturing consensus knowledge from other clients, while $P_l$ remains in the local update to extract individual characteristics. In this way, we can obtain the text feature fused with consensus knowledge and style knowledge through the attention mechanism in the encoder.

### 3.3. Visual Dynamic Adaptation

**Motivation.** Existing researches on decoupling primarily focus on different prompt settings while neglecting the diversity of vision. Unlike uniform labels, images within the same category often exhibit varied styles, leading to a diverse visual feature space. Furthermore, the alignment between image and text is influenced by the distribution of both visual and textual latent spaces. While textual prompts can optimize the textual space, the varied visual space presents difficulties in directly utilizing fixed visual features for image-text alignment. Therefore, it is essential to optimize the visual space. A straightforward way is to introduce the adapter architecture after the image encoder to reshape visual features into a new visual space.

**Dual Visual Adapters with Gate.** The adapter we employ is a compact and scalable network composed of two linear layers. Denote $W_{down} \in \mathbb{R}^{d \times r}$ as the down projection and $W_{up} \in \mathbb{R}^{r \times d}$ as the up projection, where $d$ is the dimension of encoder output and $r$ is the dimension of the hidden layer. Each linear layer is followed by a nonlinear activation layer

$\phi(\cdot)$, like RELU. Thus, an adapter can be formulated as:

$$A(x, W) = \phi(W_{up} \cdot \phi(W_{down} \cdot x)), \tag{6}$$

where $W$ is the parameter collection $\{W_{up}, W_{down}\}$. Concretely, we propose dual adapters: the general adapter $A_g$ and the specific adapter $A_s$, to reshape visual features in cooperation. Similar to decoupled prompts, the parameters of the general adapter are shared across clients to aggregate the consensus information and those of the specific adapter are updated locally to handle diverse style knowledge. Given the naive visual feature $Z_n$, the general and specific features are generated as:

$$Z_g = A_g(Z_n, W_g), \tag{7}$$
$$Z_s = A_s(Z_n, W_s). \tag{8}$$

For precise adaptation, we advocate for a dynamic combination of visual components. Driven by the Mixture of Experts (MoE) (Shazeer et al., 2017; Masoudnia & Ebrahimpour, 2014; Cai et al., 2024; Chen et al., 2023), we design a gating mechanism $G(\cdot)$ to generate weights for the feature sums. This generator is implemented by a linear layer $W_{gate}$ that maps naive visual features into weighted vectors $p \in \mathbb{R}^3$. The generated weights are calculated as follows:

$$g_x = G(Z_n, W_{gate}),$$
$$p_i(x) = \frac{\exp(g_{x_i})}{\sum_{j=1}^3 \exp\left(g_{x_j}\right)}, \tag{9}$$

where $p_i(x)$ represents the softmax-normalized weight of the $i$-th dimension of $g_x$. Specifically, $p_i(x)$ $(i \in \{1, 2, 3\})$ respectively correspond to the weights for the naive, general, and specific features, denoted as $p_n$, $p_g$ and $p_s$. After calculating the weight of each sample, the final weighted visual feature $Z_w$ is given by:

$$Z_w = p_n \cdot Z_n + p_g \cdot Z_g + p_s \cdot Z_s. \tag{10}$$

### 3.4. Language-vision Collaboration

During image-text alignment, the text encoder generates features of decoupled prompts, while dual adapters with gate reshape naive visual features produced by the image encoder. The local update follows the standard CLIP loss as Eq. (2), without any additional loss signals. The final prediction probabilities in the loss can be formulated as:

$$q(k|x) = \frac{\exp(sim(Z_w, T(P_{i,k}))/\tau)}{\sum_{c=1}^K \exp(sim(Z_w, T(P_{i,c}))/\tau)}. \tag{11}$$

After local training, the server shares the global parameters, consisting of $P_g^t$ and $W_g^t$ in our method:

$$P_g^t = \sum_{i \in C_t} \frac{m_i}{\sum_{j \in C_t} m_j} P_{g,i}^{t-1}, \tag{12}$$

**Algorithm 1** FedDDA

**Input:** Communication rounds $T$, participant set $M$, $i$-th client's private dataset $D_i$ and parameter collections $\theta_i = \{P_{g,i}, P_{l,i}, W_{g,i}, W_{s,i}, W_{gate,i}\}$, learning rate $\eta$ and tokenized embedding of guidance words GW.

**for** $t = 1, 2, ..., T$ **do**
  *Participant Side*
  **for** $i = 1, 2, ..., M$ in parallel **do**
    $P_{g,i}^t, W_{g,i}^t \leftarrow$ **Local Update**$(P_g^{t-1}, W_g^{t-1})$
  **end**

  *Server Side*
  $P_g^t = \sum_{i \in C_t} \frac{m_i}{\sum_{j \in C_t} m_j} P_{g,i}^{t-1}$
  $W_g^t = \sum_{i \in C_t} \frac{m_i}{\sum_{j \in C_t} m_j} W_{g,i}^{t-1}$
**end**

**Local Update**$(P_g, W_g)$:
$P_{g,i} = P_g$
$W_{g,i} = W_g$
**for** $(x, y) \in D_i$ **do**
  /* Textual Prior Decoupling */
  $P_i = [\boldsymbol{P_{g,i}}, \text{CLASS}, \text{GW}, \boldsymbol{P_{l,i}}]$

  /* Visual Dynamic Adaptation */
  $Z_n = I(x)$
  $Z_g = A_g(Z_n, W_{g,i}), \ Z_s = A_s(Z_n, W_{s,i})$
  $p_n, p_g, p_s \leftarrow (Z_n, W_{gate,i})$ in Eq. (9)
  $Z_w = p_n \cdot Z_n + p_g \cdot Z_g + p_s \cdot Z_s$

  /* Language-vision Collaboration */
  $\mathcal{L}_{ce} = -\sum_{k=1}^{K} y \log \frac{\exp(sim(Z_w, T(P_{i,k}))/\tau)}{\sum_{c=1}^{K} \exp(sim(Z_w, T(P_{i,c}))/\tau)}$
  $\theta_i = \theta_i - \eta \nabla \mathcal{L}_{ce}$
**end**
Return $P_{g,i}$ and $W_{g,i}$ to Server

$$W_g^t = \sum_{i \in C_t} \frac{m_i}{\sum_{j \in C_t} m_j} W_{g,i}^{t-1}. \qquad (13)$$

Through these fine-grained designs, our method implements a system of functional separation within visual-language space. In the text branch, decoupled prompts concentrate solely on textual semantics. In the visual branch, dual adapters serve as feature extractors to capture complex visual information from the naive visual features, with the proportion of generated features determined by the gate. Through this separation of functions, the components pay attention to their own roles, thereby enhancing the overall ability. The overall federated learning algorithm is shown in Algorithm 1.

**Discussion and Limitation.** Existing methods predominantly achieve decoupling through collaboration among independent prompts. However, these approaches not only compromise the semantic knowledge of the textual prompt but also overlook visual diversity. In our FedDDA, we construct prompts as Eq. (5) to fully leverage prior knowledge for effective decoupling. Additionally, unlike previous personalized methods, which restrict additional parameters exclusively to the text branch, our method introduces a vi-

sual adaptation module that dynamically adjusts the visual space for each client. The visual components inevitably introduce more training parameters, leading to increased computational and communication burden. It should be noted that this limitation is not unique to our approach but is an inherent aspect of PEFT shared across many methods. Through fine-grained design, we maximize the utility of these parameters to achieve better performances.

## 4. Experiment

### 4.1. Experimental Setup

**Datasets.** We extensively evaluate our method on the following four multi-domain classification tasks:
- Office31 (Saenko et al., 2010) contains 31 classes of common objects in office scenarios across 3 domains: Amazon (A), Webcam (W), and DSLR (D).
- PACS (Li et al., 2017) includes 4 domains: Art-painting (A), Cartoon (C), Photo (P), and Sketch (S), with 7 classes.
- OfficeHome (Venkateswara et al., 2017) consists of 4 domains: Art (A), Clipart (C), Product (P), and Real world (R), each with 65 categories.
- DomainNet (Peng et al., 2019) includes 6 domains: Clipart (C), Infograph (I), Painting (P), Quickdraw (Q), Real (R), and Sketch (S), each with 345 categories. To enhance evaluation efficiency, we choose the first 100 categories of every domain as the overall dataset.

**Data Heterogeneity.** To simulate domain shifts, we evenly assign each client a distinct domain from the dataset. We consider two client configurations: ① the client size equals that of domains; ② the client size is twice that of domains. To further evaluate performance under data heterogeneity, we take label shifts into consideration. Specifically, we partition the data within each domain based on a Dirichlet distribution (Lin et al., 2020; Kotz et al., 2004) under ② client setting, and adjust the $\beta$ parameter to model varying degrees of unbalanced label sampling.

**Counterparts.** We compare our FedDDA against two generalized approaches (Guo et al., 2023b; Li et al., 2020b) as well as two popular solutions for personalized performance (Li et al., 2024; Cui et al., 2024):
- PromptFL [TMC'23] (Guo et al., 2023b): The first to integrate prompt tuning into federated setting by learning a unified textual prompt.
- PromptFL+Prox [MLSys'20] (Li et al., 2020b): Restrict updated prompts by proximal term instead of aggregation.
- FedOTP [CVPR'24] (Li et al., 2024): Employ unbalanced optimal transport to align local visual features with global and local prompts.
- FedPGP [ICML'24] (Cui et al., 2024): Incorporate a low-rank adaptation term with an additional contrastive loss to balance generalization and personalization.

*Table 1.* **Comparison with the state-of-the-art solutions** on Office31, PACS, OfficeHome, and DomainNet tasks with domain shifts under ① and ② client settings. AVG denotes average accuracy calculated on all domains and best in **bold**. Please see details in Sec. 4.3.

| Methods | Office31 | | | | PACS | | | | | OfficeHome | | | | | DomainNet | | | | | | |
|---|---|---|---|---|---|---|---|---|---|---|---|---|---|---|---|---|---|---|---|---|---|
| | A | W | D | AVG | A | C | P | S | AVG | A | C | P | R | AVG | C | I | P | Q | R | S | AVG |
| *① One domain for one client* | | | | | | | | | | | | | | | | | | | | | |
| Zero-shot | 81.14 | 72.45 | 74.05 | 75.88 | 97.22 | 99.06 | 99.86 | 88.16 | 96.08 | 84.30 | 66.28 | 89.06 | 89.66 | 82.33 | 71.93 | 53.30 | 65.73 | 13.57 | 83.49 | 66.46 | 59.08 |
| PromptFL | 88.90 | 87.55 | 94.30 | 90.25 ↑14.37 | 98.72 | 99.23 | 99.88 | 94.81 | 98.16 ↑2.08 | 86.94 | 75.76 | 94.32 | 93.59 | 87.65 ↑5.32 | 86.55 | 70.29 | 79.89 | 34.31 | 91.54 | 79.97 | 73.76 ↑14.68 |
| PromptFL+Prox | 89.22 | 89.80 | 93.04 | 90.68 ↑14.80 | 98.76 | 99.19 | 99.88 | 94.24 | 98.02 ↑1.94 | 86.16 | 76.28 | 94.25 | 93.59 | 87.57 ↑5.24 | 87.47 | 71.25 | 82.15 | 32.63 | 91.79 | 81.20 | 74.41 ↑15.33 |
| FedOTP | 85.73 | 94.69 | 94.94 | 91.79 ↑15.91 | 98.74 | 99.43 | 99.77 | 95.55 | 98.37 ↑2.29 | 79.71 | 76.24 | 92.18 | 87.10 | 83.81 ↑1.48 | 86.73 | 69.80 | 82.05 | 50.37 | 90.75 | 82.69 | 77.06 ↑17.98 |
| FedPGP | 89.04 | 95.10 | 96.96 | 93.70 ↑17.82 | 99.16 | 99.59 | 99.87 | 95.65 | 98.57 ↑2.49 | **88.55** | 77.20 | 95.06 | **93.77** | 88.65 ↑6.32 | **89.79** | **77.68** | 87.91 | 53.08 | 93.92 | 86.50 | 81.48 ↑22.40 |
| FedDDA | **89.32** | **97.14** | **98.23** | **94.90** ↑19.02 | **99.34** | **99.74** | **99.88** | **96.65** | **98.90** ↑2.82 | 87.07 | **78.67** | **96.32** | 93.52 | **88.89** ↑6.56 | 89.74 | 77.34 | **88.46** | **63.67** | **94.14** | **88.06** | **83.57** ↑24.49 |
| *② One domain for two clients* | | | | | | | | | | | | | | | | | | | | | |
| Zero-shot | 81.14 | 72.66 | 73.99 | 75.93 | 97.22 | 99.06 | 99.86 | 88.16 | 96.08 | 84.33 | 66.28 | 89.07 | 89.69 | 82.34 | 71.94 | 53.31 | 65.73 | 13.57 | 83.49 | 66.46 | 59.08 |
| PromptFL | **89.89** | 84.89 | 92.19 | 88.99 ↑13.06 | 98.69 | 99.22 | 99.88 | 94.86 | 98.17 ↑2.09 | 86.65 | 75.38 | 94.51 | **93.49** | 87.51 ↑5.17 | 86.57 | 71.29 | 81.90 | 35.34 | 92.40 | 80.3 | 74.63 ↑15.55 |
| PromptFL+Prox | 89.25 | 86.85 | 91.80 | 89.30 ↑13.37 | 98.47 | 99.16 | 99.83 | 94.55 | 98.00 ↑1.92 | 86.13 | 74.60 | 94.40 | 93.44 | 87.15 ↑4.81 | 86.48 | 72.63 | 81.72 | 34.43 | 92.31 | 80.85 | 74.73 ↑15.65 |
| FedOTP | 84.73 | 89.16 | 95.94 | 89.94 ↑14.01 | 98.43 | 99.39 | 99.75 | 94.91 | 98.12 ↑2.04 | 77.96 | 73.96 | 91.20 | 87.10 | 82.56 ↑0.22 | 85.73 | 69.50 | 81.45 | 48.49 | 90.38 | 82.11 | 76.27 ↑17.19 |
| FedPGP | 89.24 | 91.66 | 95.18 | 92.03 ↑16.10 | 98.96 | 99.60 | 99.94 | 95.39 | 98.47 ↑2.39 | **87.92** | 77.01 | 94.96 | 92.71 | 88.15 ↑5.81 | **88.89** | **77.37** | 87.08 | 51.14 | **93.70** | 86.07 | 80.71 ↑21.63 |
| FedDDA | 88.39 | **95.06** | 98.10 | 93.85 ↑17.92 | **99.13** | **99.77** | **99.89** | **95.96** | **98.69** ↑2.61 | 86.69 | **77.98** | **95.27** | 92.96 | **88.22** ↑5.88 | 88.65 | 76.07 | **86.67** | **61.76** | 93.47 | **86.80** | **82.24** ↑23.16 |

*Table 2.* **Comparison with the state-of-the-art solutions** on Office31, PACS, OfficeHome, and DomainNet with domain shifts and label shifts under ② client setting. Refer to Sec. 4.3.

| Methods | Office31 | | | PACS | | | OfficeHome | | | DomainNet | | |
|---|---|---|---|---|---|---|---|---|---|---|---|---|
| | $\beta = 0.1$ | $\beta = 0.3$ | $\beta = 0.5$ | $\beta = 0.1$ | $\beta = 0.3$ | $\beta = 0.5$ | $\beta = 0.1$ | $\beta = 0.3$ | $\beta = 0.5$ | $\beta = 0.1$ | $\beta = 0.3$ | $\beta = 0.5$ |
| Zero-shot | 75.36 | 75.01 | 75.49 | 95.97 | 96.01 | 95.98 | 82.27 | 82.39 | 82.30 | 59.07 | 59.24 | 59.11 |
| PromptFL | 89.01 ↑13.65 | 90.44 ↑15.43 | 88.40 ↑12.91 | 97.57 ↑1.60 | 98.02 ↑2.01 | 97.98 ↑2.00 | 87.35 ↑5.08 | 87.23 ↑4.84 | 87.01 ↑4.71 | 73.14 ↑14.07 | 73.88 ↑14.64 | 74.23 ↑15.12 |
| PromptFL+Prox | 89.27 ↑13.91 | 89.66 ↑14.65 | 88.11 ↑12.62 | 97.94 ↑1.97 | 98.01 ↑2.00 | 98.05 ↑2.07 | 87.46 ↑5.19 | 87.32 ↑4.93 | 87.36 ↑5.06 | 73.41 ↑14.34 | 73.66 ↑14.42 | 74.07 ↑14.96 |
| FedOTP | 90.78 ↑15.42 | 90.54 ↑15.53 | 88.75 ↑13.26 | 98.87 ↑2.90 | 98.62 ↑2.61 | 98.49 ↑2.51 | 84.81 ↑2.54 | 85.66 ↑3.27 | 84.28 ↑1.98 | 77.52 ↑18.45 | 77.70 ↑18.46 | 76.94 ↑17.83 |
| FedPGP | 91.78 ↑16.42 | 90.88 ↑15.87 | 91.68 ↑16.19 | 99.10 ↑3.13 | 98.82 ↑2.81 | 98.72 ↑2.74 | 89.49 ↑7.22 | 89.63 ↑7.24 | 88.78 ↑6.48 | 80.72 ↑21.65 | 82.46 ↑23.22 | 82.12 ↑23.01 |
| FedDDA | **94.68** ↑19.32 | **94.34** ↑19.33 | **94.73** ↑19.24 | **99.39** ↑3.42 | **99.06** ↑3.05 | **99.10** ↑3.12 | **89.85** ↑7.58 | **90.25** ↑7.86 | **89.24** ↑6.94 | **82.98** ↑23.91 | **83.24** ↑24.00 | **82.82** ↑23.71 |

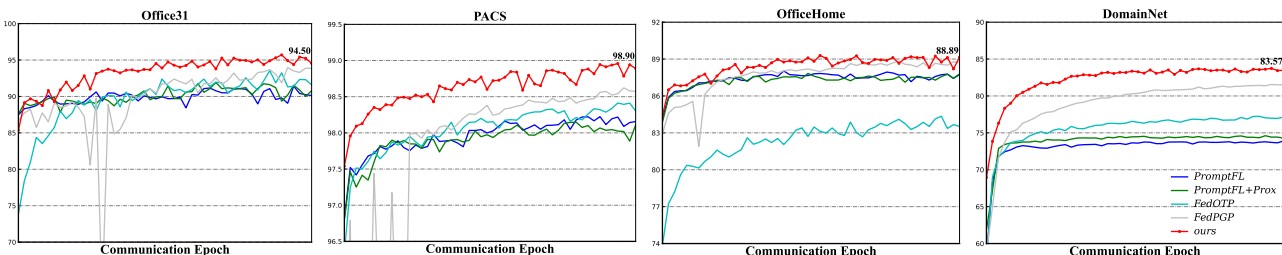

*Figure 3.* **Comparison of convergence of average accuracy with the SOTA methods** on Office31, PACS, OfficeHome, and DomainNet tasks with domain shifts under ① client setting. Refer to Sec. 4.3.

**Implementation Details.** For a fair comparison, following (Guo et al., 2023b; Li et al., 2024; Zhang et al., 2023; Huang et al., 2023), we use the same settings across all experiments. We use the publicly available CLIP (Radford et al., 2021) model with the ViT-B/16 as the backbone model. The prompt length is set to 16 and the prompts are randomly initialized with the normal distribution. We utilize SGD optimizer (Robbins & Monro, 1951) to optimize selected candidate parameters for 50 communication rounds with 1 local epoch. The learning rate $lr$ is 0.001 and the train batch size of images is 32. We fixed the random seed at 1 to ensure reproduction.

### 4.2. Ablation Study

For thorough assessment, we perform a set of ablation studies to investigate the efficacy of each component under multi-domain scenarios. To improve the efficiency of comparison, the experiments are set up on two datasets, Office31 and PACS, under ② client setting.

**Efficacy of Textual Prior Decoupling.** To further explore the impact of different prompt settings, we conduct experiments without visual adaptation module, as shown in Fig. 4. Notably, the case with a single global prompt has 32 tokens to eliminate the difference that comes from the number of updated parameters. The results indicate that a global prompt is insufficient to match diverse distributions across clients, resulting in a lower performance compared to decoupled prompts consisting of global and local parts. Furthermore, with the help of guidance words, textual decoupling achieves higher accuracy, manifesting that leveraging prior knowledge can channel more stylized information. Notably, we observe that the benefits of guidance words are smaller than those gained from the decoupled prompts. The reason is that visual diversity hinders prompts from focusing on semantic decoupling so that guidance words only capture weak semantic knowledge.

**Efficacy of Visual Dynamic Adaptation.** Through detailed experiments, we investigate the capabilities of three visual components: the general adapter, the specific adapter, and

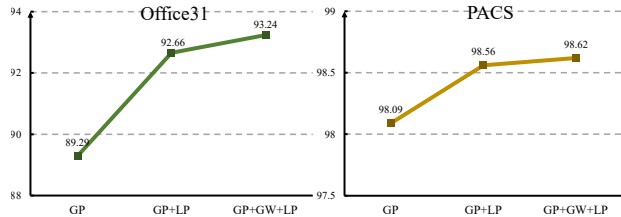

*Figure 4.* **Comparison of different prompt designs** in Office31 and PACS tasks. We present the change of accuracy from only global prompts (GP) to the decoupled prompts (GP+LP), and finally to the decoupled prompts connected by guidance words (GP+GW+LP) as shown in Eq. (5). See details in Sec. 4.2.

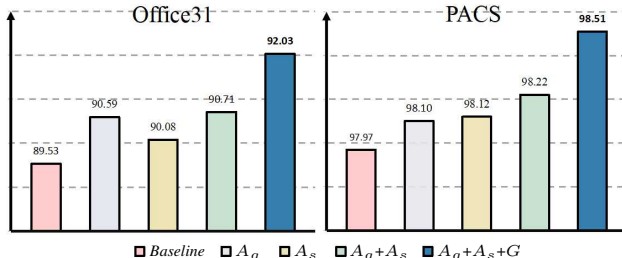

*Figure 5.* **Analysis of visual components** in Office31 and PACS tasks, including general adapter ($A_g$ in Eq. (7)), specific adapter ($A_s$ in Eq. (8)) and gate (G in Eq. (9)). See details in Sec. 4.2.

the gate mechanism. To ensure a fair comparison, we set the prompt as shared vectors. We consider five cases for the final visual feature: (1) naive visual feature ($Z_n$); (2) the average of naive and general features ($\frac{Z_n+Z_g}{2}$); (3) the average of naive and specific features ($\frac{Z_n+Z_s}{2}$); (4) the average of naive, general and specific features ($\frac{Z_n+Z_g+Z_s}{3}$); (5) the weighted sum of naive, general and specific features (Eq. (10)). As seen in Fig. 5, the incorporation of general or specific adapters achieves better performance than naive learning, validating the efficacy of the adapter architecture. This also evidences that both consensus and style knowledge contribute to the adaptation of visual space. Moreover, we explore the combination of general and specific adapters, which obtains greater discernment than using either adapter alone. Furthermore, the inclusion of the gate mechanism significantly increases performance, supporting our motivation for dynamic adaptability on multiple domains.

**Efficacy of Collaboration.** Note that the textual decoupled prompts learn fine-grained representations, and that dual visual adapters with gate reshape visual features based on client specifics. We present the ablation study of two modules in Tab. 3. The final results illustrate that both Textual Prior Decoupling and Visual Dynamic Adaptation contribute significantly to the performance of the model. The combination of two modules yields the best results, underscoring the effectiveness of our work.

*Table 3.* **Ablation study of** Textual Prior Decoupling (TPD in Sec. 3.2) and Visual Dynamic Adaptation (VDA in Sec. 3.3) in Office31 and PACS tasks. Please see details in Sec. 4.2.

| TPD | VDA | Office31 | | PACS | |
| --- | --- | --- | --- | --- | --- |
| | | AVG | Δ | AVG | Δ |
| | | 89.53 | - | 97.97 | - |
| ✓ | | 93.24 | +3.71 | 98.62 | +0.65 |
| | ✓ | 92.03 | +2.50 | 98.51 | +0.54 |
| ✓ | ✓ | **93.85** | **+4.32** | **98.69** | **+0.72** |

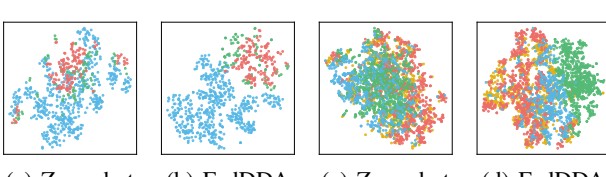

(a) Zero-shot    (b) FedDDA    (c) Zero-shot    (d) FedDDA

*Figure 6.* **t-SNE Visualization of Visual Space** on participants from Office31 (a) (b) and OfficeHome (c) (d). Refer to Sec. 4.3.

### 4.3. Comparison to State-of-the-Arts

The results in Tab. 1 plot the final accuracy against state-of-the-art (SOTA) methods on Office31, PACS, OfficeHome, and DomainNet with **domain shifts** under two types of client settings. Among most domains, our work achieves the highest accuracy, confirming that FedDDA can effectively boost performance across different domains. As shown in Fig. 3, the accuracy curve reveals our method's high accuracy and fast convergence across the four datasets. We visualize the t-SNE visualization analysis of visual space in Fig. 6. We further conduct the experiments on datasets with **domain shifts** and **label shifts** in Tab. 2. Specifically, the division of domains among clients remains unchanged as ②, while clients within the same domain are assigned data partitioned by Dirichlet strategy. We set the parameter $\beta$ in {0.1, 0.3, 0.5} to simulate varying levels of label shifts. FedDDA achieves the highest accuracy across all settings, ensuring the robustness and compatibility of our work. These comprehensive experiments illustrate that our method significantly outperforms its counterparts, highlighting the effectiveness of our design under data heterogeneity.

## 5. Conclusion

In this paper, we explore the data heterogeneity problem in FedPEFT. Our work introduces a simple yet effective algorithm, FedDDA. We leverage Textual Prior Decoupling to learn fine-grained representations for robust language supervision and introduce Visual Dynamic Adaptation to reshape visual feature space dynamically. The effectiveness of FedDDA has been thoroughly validated with popular counterparts over various classification tasks. We wish this work to pave the way for future research on FedPEFT.

**Acknowledgement.** This work is supported by National Natural Science Foundation of China under

Grant (62361166629, 62176188, 623B2080), the National Key Research and Development Program of China (2024YFC3308400), Postdoctoral Fellowship Program of China Postdoctoral Science Foundation (GZC20241268, 2024M762479), and the Wuhan University Undergraduate Innovation Research Fund Project. The supercomputing system at the Supercomputing Center of Wuhan University supported the numerical calculations in this paper.

## Impact Statement

This paper presents work whose goal is to advance the field of Machine Learning. There are many potential societal consequences of our work, none of which we feel must be specifically highlighted here.

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
