# OpenReview forum: "Federated Disentangled Tuning with Textual Prior Decoupling and Visual Dynamic Adaptation"
_ICML.cc/2025/Conference — ICML 2025 poster_

### Official Review · Reviewer_UBUx · 2025-03-05

**Overall Recommendation:** 4

**Summary:**

This paper proposes FedDDA, a federated framework designed to enhance the adaptation of Vision-Language Models under data heterogeneity. Toward existing works, the authors highlight two key limitations in both textual and visual modality: Textual Property Loss and Visual Feature Diversity. To address these challenges, this work proposes Textual Prior Decoupling (TPD), which learns fine-grained representations. Moreover, it proposes Visual Dynamic Adaptation (VDA) to optimize the visual space to align with the textual space. The method is evaluated across four classification tasks, demonstrating robustness and generalizability.

**Claims And Evidence:**

W1. The proposal of dual adapters is confusing. I can understand the need to reshape visual features for visual diversity. However, the rationale for employing both global and local adapters is unclear.

**Essential References Not Discussed:**

N/A

**Experimental Designs Or Analyses:**

S1. The paper provides extensive empirical validation through experiments on multiple scenarios.

W2. The results on DomainNet, especially in the Quickdraw domain, are notably promising compared to the other three datasets. The authors should clarify the reasons behind these results.

W3. Reducing the computational and communication demands is the foundational goal of FedPEFT. In Section 3.4, the authors only discuss the increased burden from a conceptual standpoint. An analysis backed by empirical results is expected.

**Methods And Evaluation Criteria:**

This paper introduces personalized components from both modalities, which make sense for adapting VLM to downstream tasks in a federated setting. Additionally, the average accuracy effectively demonstrates the effectiveness of the proposed method.

**Other Comments Or Suggestions:**

N/A

**Other Strengths And Weaknesses:**

S2. This paper addresses the challenge of data heterogeneity in the downstream adaptation of VLM under federated settings, which are crucial for applications in the real world. It facilitates personalized adaptation using distributed data while preserving privacy.

S3. The paper is well-structured and mostly clearly written. The authors divide the problem into two parts based on both modalities, presenting a clear flow that enhances comprehension.

**Questions For Authors:**

I am willing to consider raising the score based on your rebuttal to the following questions:

Q1. The rationale for employing dual adapters should be clarified.

Q2. Why does the performance on DomainNet appear more promising compared to the other three datasets?

Q3. Reducing the computational and communication demands is the foundational goal of FedPEFT. Can the authors provide more empirical details on the associated costs?

**Relation To Broader Scientific Literature:**

The core idea of existing personalized works [1][2][3] is to decouple prompts into independent components with specific properties (e.g., global and local). This paper figures out two key limitations of these works and designs targeted modules to tackle.

[1] Global and local prompts cooperation via optimal transport for federated learning, CVPR2024

[2] Harmonizing generalization and personalization in federated prompt learning, ICML2024

[3] Diprompt: Disentangled prompt tuning for multiple latent domain generalization in federated learning, CVPR2024

**Theoretical Claims:**

It’s easy to follow the methods with precise theoretical definitions and clear pipelines.

---

> ### Author Rebuttal · Authors · 2025-03-31
>
> Dear Reviewer UBUx:
>
> Thank you for your positive response! We value the constructive feedback and have carefully considered the concerns raised. Below we provide detailed responses to address the reviewer's concerns:
>
> **Weakness & Question**
>
> **W1 & Q1: The rationale for employing dual adapters should be clarified.**
>
> **A1:**  We apologize for the confusion. In the vision branch, we tend to introduce the adapter architecture to reshape image features. We observe that both global and local adapters contribute to optimizing the visual feature space, thereby enhancing vision-language alignment. Building on this insight, we explore their combination (dual adapters) to enable mutual reinforcement between consensus and style knowledge. This design takes full account of the nature of federated learning, helping to preserve the generalization capability while improving personalized representation.
>
> **W2 & Q2:  Why does the performance on DomainNet appear more promising compared to the other three datasets?**
>
> **A2:**  Thank you for your detailed observation. Compared to the other three datasets, DomainNet is significantly larger and exhibits greater domain diversity. This characteristic poses a greater challenge of visual feature diversity, which is the primary focus of our study. With targeted visual components (dual adapters with gate), our approach is more compatible with handling such complex scenarios compared to other methods. This is clearly demonstrated by its superior performance on DomainNet.
>
> **W3 & Q3:  The empirical record on the associated costs.**
>
> **A3:** Thank you for this valuable suggestion. We measure the communication cost by the size of uploaded data per round per client, and computation cost by the number of floating point operations (FLOPs) in the same batch. Below we present a comparison of both communication and computation costs.
>
> Table 2. Comparison of communication cost (KB) and computation cost (G).
>
> | Method                    | Communication Cost  (KB) | Computation Cost (G) |
> | ------------------------- | :----------------------- | -------------------- |
> | PromptFL                  | 16.00                    | 1160.33              |
> | PromptFL+Prox ($\mu=0.5$) | 16.00                    | 1160.33              |
> | FedOTP                    | 16.00                    | 1247.02              |
> | FedPGP                    | 16.00                    | 1323.26              |
> | FedDDA (ours)             | 272.00                   | 1160.38              |
>
> The introduction of the general adapter in our method adds more global parameters, inevitably leading to an increased communication burden. Meanwhile, the local updates in our method follow the standard CLIP loss and involve only one combined prompt that needs to be encoded. As a result, the overall computational cost of our approach remains comparable to that of PromptFL.

---

> > ### Comment · Reviewer_UBUx · 2025-04-03
> >
> > Thank the authors for providing further experimental analysis and clarification on the rationale behind dual adapters, which have addressed most of my questions and concerns. I appreciate this manuscript for its interesting task, clear presentation, and extensive experiments. To sum up, I will raise my score accordingly.

---

### Official Review · Reviewer_287K · 2025-03-10

**Overall Recommendation:** 3

**Summary:**

The paper proposes a novel FedDDA approach to improve federated learning for fine-tuning Vision-Language Models (VLMs) under data heterogeneity. The focus is on addressing two primary challenges in both language and vision modality. On the textual side, the method integrates two prompts (global and local) in a single part to overcome the textual property loss existing in recent works. On the visual side, it employs a dual-adapter architecture with a gate network to mitigate visual feature diversity. The paper demonstrates the effectiveness of FedDDA through extensive experiments on various datasets.

**Claims And Evidence:**

[weakness] Strange decoupling: The authors claim to decouple the prompt, but in practice, the two decoupled parts are merged into a single feature. This undermines the effect of the decoupling. Could the authors explain how their decoupling strategy makes sense?

**Essential References Not Discussed:**

The relevant references are comprehensive.

**Experimental Designs Or Analyses:**

[strength] Extensive experimental validation: This work provides a comprehensive performance evaluation through experiments conducted on an extensive set of scenarios, including a convincing ablation study.

**Methods And Evaluation Criteria:**

The proposed methods (introduction of personalized components) and evaluation criteria (average accuracy) align well with the problem of personalized federated learning for VLMs.

**Other Comments Or Suggestions:**

N/A

**Other Strengths And Weaknesses:**

[strength] Crucial challenge: This paper addresses a crucial challenge of personalized adaptation in federated applications of VLMs to downstream tasks, which require fine-tuning with diverse data sources.

[strength] Clear presentations: The figures are well-presented, making it easy to follow the proposed method. Additionally, the charts are clear and effective in illustrating the results.

[weakness] Unclear rationale for specific technologies: While the paper discusses various parameter-efficient fine-tuning technologies such as prompts, adapters, and LoRa, it lacks a clear explanation of why a specific technology is chosen for a particular modality. The rationale behind these choices should be made clearer, particularly in Section 2.1.

[weakness] Writing issues: Some minor writing issues need attention. For example, in Line 81, a space is missing in “… prior knowledge.II) Visual Feature Diversity …”.

**Questions For Authors:**

Please check the weakness mentioned above.

**Relation To Broader Scientific Literature:**

The authors categorize state-of-the-art methods such as FedOTP [1], FedPGP [2] as decoupled methods, where the prompt is divided into global and local parts. This paper builds upon these approaches by addressing their limitations, particularly in relation to the vision modality. Its main contribution is to fill in the neglect of vision modality in previous works.

References:\
[1] Global and local prompts cooperation via optimal transport for federated learning. Li et al. CVPR 2024.\
[2] Harmonizing generalization and personalization in federated prompt learning. Cui et al. ICML 2024.

**Theoretical Claims:**

The theoretical part focuses on the definition of the proposed architecture, providing clear insights into the framework.

---

> ### Author Rebuttal · Authors · 2025-03-31
>
> Dear Reviewer 287K:
>
> We sincerely appreciate your constructive and insightful comments. We will explain your concerns point by point.
>
> **Weakness**
>
> **W1: Strange decoupling.**
>
> **A1:** A key challenge in decoupling prompts lies in achieving efficient collaboration between prompts with different attributes. One approach is to decouple and collaborate at the feature level, like FedOTP [1] and FedPGP [2]. Instead, we aim to decouple at the semantic level, i.e., in the prompt construction phase, and facilitate collaboration through the pre-trained encoder. Our goal is to coordinate decoupled prompts by leveraging prior knowledge from the encoder to generate a feature that incorporates both consensus and style knowledge.
>
> **W2: Unclear rationale for specific technologies.**
>
> **A2:** With respect to the textual modality, replacing the textual inputs of the CLIP model with prompts is a common approach to learning task-specific knowledge. Our work continues this textual paradigm, and we introduce a visual limitation building on this paradigm: visual feature diversity. We aim to address this challenge by dynamically adapting the high-dimensional visual feature space. A straightforward strategy is to leverage adapters, which are flexible and efficient. We will explicitly state the reasoning behind selecting each PEFT method for different modalities in Section 2.1.
>
> **W3: Writing issues.**
>
> **A3:** Thank you for your tips. We have corrected the error you noted in Line 81 and other potential issues.
>
> [1] Global and local prompts cooperation via optimal transport for federated learning. Li et al. CVPR 2024.
>
> [2] Harmonizing generalization and personalization in federated prompt learning. Cui et al. ICML 2024.

---

### Official Review · Reviewer_oran · 2025-03-11

**Overall Recommendation:** 4

**Summary:**

This paper addresses the challenge of data heterogeneity in Federated Parameter-Efficient Fine-Tuning. To tackle this issue, the authors propose FedDDA, which is designed from two modalities. In the text modality, the proposed approach decouples prompts into global and local components, which are connected through guidance words. For the vision side, the method utilizes dual adapters and a gating mechanism to blend naive, consensus, and style visual features dynamically. Extensive experiments across various scenarios demonstrate the superiority of the proposed method.

**Claims And Evidence:**

- The paper introduces VDA to optimize the visual space and employs t-SNE visualization to substantiate this claim. However, the changes in the visual space, as shown in Fig. 6, appear to be relatively minor. It is unclear whether such minor variations can yield significant contributions. The authors should provide further clarification on this point.

**Essential References Not Discussed:**

No essential references have been left out.

**Experimental Designs Or Analyses:**

Strengths：
- The experiments are comprehensive, covering a wide range of scenarios to thoroughly evaluate the proposed method. Ablation studies are thoroughly conducted, providing clear insights into the contributions of each component.
Weaknesses:
- The results in Tab.1 and Tab.2 report improvement against a weak baseline rather than relatively SOTA methods.

**Methods And Evaluation Criteria:**

The proposed method addresses the targeted problem, and the evaluation criteria are reasonable.

**Other Comments Or Suggestions:**

NO

**Other Strengths And Weaknesses:**

Strengths：
- The paper fully considers the properties of both federated learning and vision-language models. The proposed method employs global and local components in both text and image branches.
- The paper is well-motivated. The authors propose two key limitations based on existing studies and further provide a targeted solution to address these limitations.
- The manuscript is well-structured, clearly delineating the proposed methodology, experimental setup, and results.
Weaknesses:
- This work employs several existing techniques, including prompts, adapters, and MoE. While this is not an issue, the rationale behind selecting each specific technique is not clearly explained. For instance, in response to Visual Feature Diversity, the authors aim to dynamically adjust the visual feature space, yet visual prompts could achieve a similar effect.

**Questions For Authors:**

Please refer to the above weaknesses.

**Relation To Broader Scientific Literature:**

Recent studies on the personalized adaptation of vision-language models in federated settings primarily focus on prompt decoupling, leveraging loss signals, or multi-stage training. In the text modality, this work refines previous decoupling strategies by integrating initially independent prompts into a single prompt, thereby preserving textual attributes. As for the visual modality, it takes visual diversity into consideration, a factor that is often overlooked in existing research.

**Theoretical Claims:**

The theoretical formula is clear and straightforward, providing an easy-to-follow structure.

---

> ### Author Rebuttal · Authors · 2025-03-31
>
> Dear Reviewer oran:
>
> Thank you very much for your recognition of our work and for raising thoughtful questions and concerns about our work. We have carefully considered each comment and provided responses.
>
> **Weakness**
>
> **W1: Analysis of t-SNE visualization.**
>
> **A1:**  Thank you for your insightful question. Unlike traditional unimodal architectures, CLIP requires alignment between visual and textual space. In our work, each participant's visual space is optimized to better align with the textual space, which is also being refined simultaneously. This bidirectional optimization leads to improved overall performance, even though the visual space itself undergoes relatively minor changes.
>
> **W2: The results in Tab.1 and Tab.2 report improvement against a weak baseline rather than relatively SOTA methods.**
>
> **A2:**  We will revise Tab. 1 and Tab. 2 to provide a more detailed comparison in the final version, not simply an improvement over the weak baseline. Thanks for your advice.
>
> **W3: The rationale behind selecting each specific technique is not clearly explained.**
>
> **A3:** For the textual modality, prompt tuning has emerged as an efficient way to learn specific semantic knowledge on downstream tasks. Therefore, recent research focuses on how to better use prompt tuning, as we do. For the visual modality, our goal is to go for dynamic adaptation of the visual space to address visual feature diversity, thus better aligning with textual space. The available methods are adapters or visual prompts. Similar to the textual prompt, we explore the combination of global and local components, which is easy to implement in adapter tuning. However, managing the trade-off between global and local components in visual prompts remains challenging, particularly in determining their placement and properties. To handle this trade-off in adapter tuning, we introduce a gating mechanism, which dynamically learns the summation weights to coordinate multiple generated features. Thank you for your constructive comments.

---

### Official Review · Reviewer_Zexp · 2025-03-14

**Overall Recommendation:** 3

**Summary:**

This paper considers the collaborative effectiveness caused by data heterogeneity during Federated Parameter-Efficient Fine-Tuning process. A simple yet effective algorithm is proposed to address the above challenges via Textual Prior Decoupling and Visual Dynamic Adaptation. In this case, both modalities can be collaborative in an unbiased manner. Experiments on various datasets demonstrate the efficiency of the proposed method.

**Claims And Evidence:**

Most of the claims made in the submission are supported by clear and convincing evidence.

**Essential References Not Discussed:**

None

**Experimental Designs Or Analyses:**

The experimental results are comprehensive, with the author conducting various experiments under different settings to ensure thorough evaluation.

**Methods And Evaluation Criteria:**

Yes

**Other Comments Or Suggestions:**

See the above weaknesses.

**Other Strengths And Weaknesses:**

Strengths

1. The paper is well-written and easy to understand. The motivation and proposed solution are reasonable and have a certain degree of novelty.
2. The authors provide a clear and well-structured discussion for the experimental part.

Weaknesses

1. It is suggested to add a more detailed comparison or analysis for the existing decoupled works in Related Work, e.g., why decoupling prompts into independent components is insufficient to achieve better collaboration? What are the differences between FedDDA and DiPrompT?
2. The intuition behind the decoupled textual prompt is not clear, why define it in a combined manner instead of an independent component?
3. It seems that DiPrompT can be also a baseline in the comparison experiments, please clarify why this method is not included in the performance comparison.
4. There lack of experiments on analysing the performance impact of the number of total clients, i.e., 50 clients or more.

**Questions For Authors:**

See the above weaknesses.

**Relation To Broader Scientific Literature:**

None

**Theoretical Claims:**

N.A. There is no proof for theoretical claims.

---

> ### Author Rebuttal · Authors · 2025-03-31
>
> Dear Reviewer Zexp:
>
> Thanks for raising thoughtful questions and concerns. We sincerely hope this point-to-point response allows the reviewer to update the score. We will add a more detailed comparison or analysis of the existing decoupled works in the final version.
>
> **Weakness**
>
> **W1: Analysis of motivation. (Why decoupling prompts into independent components is insufficient to achieve better collaboration?  Why define the decoupled in a combined manner instead of an independent component?)**
>
> **A1:** We apologize for the confusion. We define the problem with current decoupling methods as textual property loss. Specifically, after extensive training, the text encoder is capable of recognizing textual relationships between tokens—what we refer to as textual property. For instance, it tends to perform better with fixed texts like "[class] with a style of [domain]", demonstrating its ability to capture concepts such as style. However, methods that decouple prompts into independent components overlook this capability, as they process prompts in isolation while the encoder can only handle tokens that come in together. Additionally, this paradigm depends on newly designed losses to guide the optimization, which is irrelevant to textual properties. Therefore, such methods fail to learn the knowledge of textual relationships between independent prompts. We argue that textual relationship is crucial for coordinating decoupled prompts effectively. Unfortunately, current methods weaken these relationships, making them insufficient for effective collaboration.
>
> Thus, we propose a combined approach, where multiple prompts are integrated into a single prompt. Decoupled prompts are fed into the textual encoder together. It can maximize the use of the encoder's knowledge to achieve better textual collaboration. As we demonstrate below, the method that simply combines global and local prompts outperforms existing methods, validating its effectiveness.
>
> *Table: Comparison of simple combination with several methods on Office31 and PACS with $\beta=0.0$. Simple combination is denoted as $[P_g, class, P_l]$, where $P_g$ is the global prompt and $P_l$ is the local prompt.*
>
> | Method                    | Office31  | PACS      |
> | ------------------------- | :-------- | --------- |
> | PromptFL                  | 88.99     | 98.17     |
> | PromptFL+Prox ($\mu=0.5$) | 89.30     | 98.00     |
> | FedOTP                    | 89.94     | 98.12     |
> | FedPGP                    | 92.03     | 98.47     |
> | Simple Combination        | **92.66** | **98.56** |
>
> **W2: The differences between FedDDA and DiPrompT.**
>
> **A2:** Thank you for your valuable feedback. The primary difference between FedDDA and DiPrompT is whether the decoupled prompts are independent or not. This also marks the key difference between FedDDA and other previous decoupled methods. Previous decoupled methods, including DiPrompT, decouple prompts into independent components, each designed with distinct properties. These properties are typically derived from global or local federated setups and are further enhanced through external loss signals or multi-stage training. Instead, our method combines decoupled prompts into a single prompt, which maximizes the use of prior knowledge to achieve better textual collaboration.
>
> **W3: The reason why DiPrompT is not included in the performance comparison.**
>
> **A3:**  We fully understand your concern about DiPrompT as a baseline. However, DiPrompT and our FedDDA address different problems with distinct experimental setups. DiPrompT is designed for domain generalization, while our work focuses on personalized adaptation to optimize performance within each client's own domain. Unfortunately, the lack of open-source code ultimately prevents us from including this approach in our evaluation. Once it becomes available, we will make every effort to conduct a thorough comparison. Thank you for your insightful comment.
>
> **W4: Lack of experiments on analyzing the performance impact of the number of total clients.**
>
> **A4:**   We conduct the experiments on the performance impact of the number of total clients as shown in the following Table. Our method constantly achieves better results than other methods.
>
> *Table: The performance impact of the number of total clients on PACS with $\beta=0.0$.*
>
> | Client Number             | 4         | 8         | 20        | 40        | 50        |
> | ------------------------- | :-------- | --------- | --------- | --------- | --------- |
> | PromptFL                  | 98.16     | 98.17     | 98.08     | 97.98     | 97.96     |
> | PromptFL+Prox ($\mu=0.5$) | 98.02     | 98.00     | 98.03     | 97.98     | 97.96     |
> | FedOTP                    | 98.37     | 98.12     | 97.71     | 97.38     | 97.25     |
> | FedPGP                    | 98.57     | 98.47     | 98.14     | 97.61     | 97.41     |
> | FedDDA (ours)             | **98.90** | **98.69** | **98.35** | **98.08** | **98.05** |

---

> > ### Comment · Reviewer_Zexp · 2025-04-09
> >
> > After reading all the comments, I think the authors address most of my concerns. To sum up, I will raise my score accordingly.

---

### Decision · Program_Chairs · 2025-05-01

**Decision:**

Accept (poster)

**Comment:**

The paper received four accept ratings and the reviews are mostly positive without serious flaws. Reviewer Zexp found the method novel and the results comprehensive, but questioned the intuition behind some design choices. The rebuttal addressed most of these questions and the reviewer raised the score from weak reject to weak accept. Reviewer oran also appreciated the method and results but pointed out that the baselines compared in tables 1 and 2 are not the latest. The AC has checked the tables and found that the paper has included two latest works in the comparison, which are FedOTP (CVPR'24) and FedPGP (ICML'24). Reviewers 287K and UBUx were satisfied with the results and writing, but raised concerns about the rationale behind the design of using learnable prompt for the text modality while adapter for the vision modality. The rebuttal had addressed these concerns. Overall, the reviewers found this work interesting and the results sufficient for justifying the effectiveness of the method. The AC recommends that the paper be accepted.